# Waste Natural Polymers as Potential Fillers for Biodegradable Latex-Based Composites: A Review

**DOI:** 10.3390/polym13203600

**Published:** 2021-10-19

**Authors:** D. N. Syuhada, A. R. Azura

**Affiliations:** School of Materials and Mineral Resources Engineering, Engineering Campus, Universiti Sains Malaysia, Nibong Tebal 14300, Penang, Malaysia; nsyuhadadzulkafly@gmail.com

**Keywords:** extractions, cellulose, crosslink mechanism, biodegradable, waste materials, natural filler, latex, composite, mechanical properties, surface modifications

## Abstract

In recent years, biodegradable composites have become important in various fields because of the increasing awareness of the global environment. Waste natural polymers have received much attention as renewable, biodegradable, non-toxic and low-cost filler in polymer composites. In order to exploit the high potential for residual natural loading in latex composites, different types of surface modification techniques have been applied. This review discusses the preparation and characterization of the modified waste natural fillers for latex-based composites. The potency of the waste natural filler for the latex-based composites was explored with a focus on the mechanical, thermal, biodegradability and filler–latex interaction. This review also offers an update on the possible application of the waste natural filler towards the biodegradability of the latex-based composites for a more sustainable future.

## 1. Introduction

Municipal solid waste consists of everyday items such as bottles, food scraps, paints, food wrappers, packaging and newspapers are used and thrown away every day. Some of these materials are biodegradable, while others are not. Because non-biodegradable items do not decompose over time, they are frequently thrown in landfills or incinerated, creating a serious environmental concern. This is one of the most important concerns affecting society today, as the concentration of non-biodegradable plastics continue to rise at an alarming rate [1]. Conversion of industrial waste into valuable product has raised the interest in the research area and various effort is taken towards the waste to wealth concept all around the world.

Rubber or elastomer is a polymeric substance with flexibility and extensibility qualities. When a force is applied, the molecules stretch out in the direction of the applied force; when the force is released, the molecules abruptly return to their initial state. Natural rubber (NR) is a biodegradable and environmentally friendly elastomer with exceptional elasticity and outstanding characteristics under cyclical stresses [2]. NR latex is obtained from the Hevea Brasiliensis tree. NR latex is a colloidal dispersion of small polymeric particles (3 to 5 µm) covered by a layer of proteins and lipids [3]. Table 1 shows the component in the colloidal dispersion of NR latex.

NR latex is useful for a wide range of products due to the ability of the rubber particles to coalesce and form a coherent polymer layer impenetrable to air and water. The vulcanization process in the latex transforms the materials into a highly versatile raw material for the manufacture of a variety of rubber goods via dipping, molding, casting, or spreading operations. However, the film cast from NR latex is soft and tacky; thus, the latex must be prevulcanized using a sulfur, peroxide or radiation technique to give a higher grade output [4]. To achieve the elastic qualities of the rubber, the latex was treated with many chemicals that act as preservatives, anticoagulants, vulcanizing agents and antioxidants [3]. NR latex products outperform synthetic counterparts due to their excellent strength properties along with low modulus and high wet gel strength, elasticity, resilience, heat dissipation and abrasion resistance characteristics which cannot easily be mimicked by synthetic polymers.

Synthetic latex is primarily produced using emulsion polymerization and can be synthesised from a variety of monomers, including acrylates, styrene, vinyl acetate and butadiene [5]. The properties of synthetic rubber such as high temperature resistance, good resistance to abrasion, high strength is depending on the composition of the copolymers. Both NR and synthetic rubber are different in microstructure but contain isoprene as the main chain [3]. Even though NR latex products can biodegrade in the soil, the process is slow due to the large molecular weight of NR latex macromolecules with additional chemical crosslinking that formed through the vulcanization process. These cross-linkages are stable under ambient conditions and can withstand the soil environment for a prolonged period of time. Another factor that may conduce to the delayed degradation rate is the use of additives such as antioxidants in vulcanized NR latex, which is intended to limit microbial growth upon disposal [6].

Currently, the majority of rubber waste is burned or disposed of in landfills. Microbes play a role in the biodegradation process by releasing extracellular enzymes that cleave polymer chains into tiny molecules, which are then absorbed into the microorganism’s cell to be used as a source of carbon and energy. CO_2_, H_2_O and other metabolic products are released at the end of the process and can be utilized by other living creatures. Therefore, biodegradation is another way of degrading rubber waste that has the potential to resolve environmental problems. Microbial breakdown of polyisoprene rubber, on the other hand, is a very sluggish process that can take months or even years [7]. The addition of compounding ingredients such as reinforcing fillers, accelerators and sulfur increased the resistance of the latex towards microbial degradation caused by the decreased of access for microorganism to the rubber matrix [8].

Biodegradation is an environmentally benign alternative to conventional disposal methods in which microorganisms biologically break down complex organic chemicals in commercial items into cell biomass and less complex molecules, as well as water and either carbon dioxide or methane. For microbes to use polymers as a nutrition source, polymers must first be oxidized through an abiotic process, such as exposure to ultraviolet (UV) irradiation, heat and/or chemicals in the environment. Some microbes, such as rubber degraders, can commence the oxidation process on their own, with secreted enzymes causing polymer biofragmentation followed by bioassimilation of small cleavage pieces [3].

The global demand for natural source NR rubber gloves has been growing in recent years. Due to an increased in NR glove consumption, the formation of a significant volume of NR latex-based solid waste is unavoidable. The use of chemical agents such as fillers, antioxidants, accelerators, vulcanizing agents and inhibitors during the manufacturing process have a significant impact on the chemistry of NR latex, decreasing its inherent degradability. An excessive amount of solid waste has been generated due to increased consumption of NR latex products and their low degradation, which has negatively impacted the sustainability of the environment. The development of biodegradable NR latex composites has been widely investigated in order to reduce the environmental issues caused by rubber waste disposal [6,9,10,11,12].

Natural polymer composites made from filler have received much attention in recent years due to the benefits of natural filler and environmental concerns. In order to reduce the greenhouse effect, the composite material based on renewable and biodegradable materials is progressively replacing the synthetic fiber reinforced composite. The strength of a natural fiber composite is determined not only by the interfacial strength, but also by the fiber’s basic strength. The fiber quality is determined by the crop’s location and climate, as well as the fiber age, plant species, transit mode, storage time and inventory condition. Chemical treatments of natural fibers are used to change the physical, mechanical and thermal properties of composites to address this problem. Chemical treatments may reduce the hydrophilicity of plant fibers, clean the surface of fibers, improve the roughness of fibers and reduce the moisture content of fibers, resulting in better reinforcement–matrix interactions [13].

The growth of agricultural industries also leads to an increase in the production of biomass from the agricultural sector. The agro waste production is generated in ongoing basis and produced in large quantities. Excessive agricultural waste creates a disposal problem through waste management problems, pollution problems and also becomes a waste of primary resources. In order to deal with this problem, recycling and reusing this waste is more important. Agricultural waste is used as feedstock, biochemicals, biomaterials and also as fuel to produce heat and electricity [14]. However, due to poor fuel properties, the usage of agro waste as fuel is still minimal. Agro waste also can be beneficial if it is used properly. There are several research has been done on the agro waste as an alternative filler to synthetic filler in order to obtain renewable and environmentally friendly products. The agro waste can be generated from various sources such as date palm waste [15,16], cocoa pod husk [17], cotton [18,19,20], coconut [21,22,23], rubber wood and kenaf bast [24]. Basically, from agro waste, it contains three main constituents which is cellulose, hemicellulose and lignin.

Cellulose is the most abundant polymer produced in nature and microorganism. The major conventional resources of cellulose can be obtained from cotton and wood. The cellulose and its derivatives have been utilized for years in the industrial applications such as textile, paper and medical due to its unique properties, biodegradable and come from renewable resources [25,26]. Cellulose is polysaccharides with a chemical formula of (C_6_H_10_O_5_) _n_ composed of repeating units of β-d-glucopyranose with covalent link between the OH group of C4 and C1 carbon atoms [27]. Properties of cellulose is depending on its degree of polymerization which depends on the source of the cellulose. However, due to the high number of hydroxyl group on the glucose ring along the skeleton; thus, there is hydrogen binding between the chains. The crystallization of the chain and the presence of the two regions in cellulose, which is crystalline and amorphous, resulting the general properties of cellulose such as high strength, stiffness, durability and biocompatibility. The hydroxyl group in the cellulose gives the hypdrophilicity, chirality and biodegradability to the cellulose; in addition, these hydroxyl groups allow the chemical modification of the cellulose [28]. However, the characteristics of cellulose are highly dependent on the plant sources and the extraction process [29,30]. There are few types of cellulose that are extracted from natural resources, which are microcrystalline cellulose (MCC), cellulose nanowhiskers (CNW) and cellulose nanofibers (CNF). MCC has the properties of biodegradable, insoluble and biopolymer [31]. CNW is a needle-like cellulose, having at least 1 dimension equal or less than 100 nm and highly crystalline. CNF can be divided into two categories which are nanofibrillated cellulose and microfibrillated cellulose [29]. CNF is a long, flexible nano-string consist of alternation crystalline and amorphous. CNF has a large surface to volume ratio, high mechanical properties and ability to form a highly porous mesh [32].

The purpose of this review is to provide a summary of recent studies on the effect of incorporating waste natural filler into latex-based composites. This review also covers the extraction method and properties of waste natural filler, surface modification of waste natural filler and the preparation of latex-based composites. This review paper also discusses the potential applications of waste natural filler.

## 2. Type of Waste Natural Filler and the Extraction Methods

Various methods are used to extract the natural cellulose from plant. The most commonly used method is an acid hydrolysis method and alkaline method. The aim of these processes is to completely remove the amorphous region, for example, lignin and hemicellulose, to obtain the cellulose crystalline region which may affect the properties of the cellulose [18].

The technique involves in the extraction of cellulose from waste natural filler, including the chemical method, mechanical method and a combination of mechanical and chemical method. Chemical treatment usually involves the use of alkaline [33,34,35,36,37,38], acid hydrolyzed [39,40,41,42] and bleaching of the natural filler. Alkaline treatment ruptures the intramolecular crossed bond between lignin and hemicellulose. This process increases the filler porosity and allows the access of the alkali to the lignin which allow intensive dissolution of lignin structure [43]. The commonly use alkaline treatment for the natural filler is sodium hypochlorite [33,34,35,36] and sodium hydroxide (NaOH) [37,38]. Since the alkaline treatment mostly remove the lignin structure, this treatment is usually used as pre-treatment in the isolation of cellulose. The usage of the acid hydrolysis technique is the most common technique found to isolate nanocellulose after the pre-treatment process. The usage of acid includes hydrochloric acid [39], oxalic acid [44], formic acid [42] and sulfuric acid [41]. The drawback of using acid treatment is a large number of effluents generated during washing the filler, corrosion hazard of the acid, low thermal stability for the sulfuric acid and generate a low yield of cellulose due to degradation of cellulose [32].

A part of chemical treatment, mechanical treatment has also been used as a method to isolate nanocellulose. The example of the mechanical methods to produce nanocellulose are ball milling [37], ultrasonication [45] and homogenization [46]. High pressure and high intensity ultrasound are found to be effective mechanical treatment to produce cellulose. High-pressure homogenization generates a variety of disruptive forces that can cause the structural organization of cellulosic materials to partially dissolve, including cavitation, turbulence and shear effects [46]. Meanwhile, high-intensity ultrasonic treatments may also degrade cellulosic materials by producing strong cavitation stresses [45]. The combined technique of using mechanical and chemical treatment is called as mechano-chemical treatment. The mechanochemical technique is applied by Song et al. (2018) [40] in order to use a mild acid to isolate the cellulose. The usage of mild acid aims to reduce the degradation of cellulose, which impact the yield of the cellulose. However, mild acid needs more reaction time in order to isolate the cellulose; hence, it is assisted with mechanical forces in order to reduce the reaction time.

In order to determine the crystallinity and the crystallite size of the natural filler, x-ray diffraction (XRD) is generally used. XRD analysis is used to compare the effect of the treatment towards the improvement of the crystallinity of the natural filler, which helps to comprehend the final properties of the composites. The crystallinity index measures the difference of intensity of the lattice’s diffraction with the intensity diffraction of the amorphous. Meanwhile, the crystallite size can be measured by following Scherrer’s equation [47]. Most studies showed that the crystallinity of waste natural filler subjected to the treatment or extraction will be improved. The amorphous peak of the waste natural filler will be reduced, while the crystalline peak will be increased. Thambiraj and Ravi Shankaran [18] mentioned that the plane reflection of the original industrial waste cotton is wide and round, while after treatment, the curve becomes more sharper and narrower, which indicated the removal of lignin and hemicellulose, thus resulting the increase in crystallinity. The increase in crystallinity was related to the removal of the amorphous constituent and the arrangement of the crystalline region into more ordered structure [17].

Another important characterization of the modified waste natural filler is the morphology of the waste natural filler. The morphology of the filler is usually determined by scanning electron microscopy (SEM) and transmission electron microscopy (TEM). Using this test, the structure of the filler can be obtained, including the estimated length and width of the filler. The structure and the size of the modified waste natural filler depends on the sources of the waste natural filler and the treatment process. The structure of the treated waste natural filler will turn into a needle-like shape [24], rod-like shape [17,18,20,48], skeletal rod-like microfibril structure [15] and also acicular structure [49], as shown in Figure 1. Basically, the size of the treated waste natural filler may be reduced after treatment, resulting in the separation of individual fibers due to the removal of the cementing materials, which are lignin and hemicellulose [21]. Table 2 summarized the results from different sources of natural waste filler and the extraction method used.

However, in comparison to synthetic fibers, these natural cellulosic fibers have poor mechanical capabilities and their compatibility with the composite matrix should be improved. Furthermore, most hydroxyl and other polar groups increase the moisture absorption of natural fibers, making it more difficult to make composites with hydrophobic polymer matrices). As a result, natural fibers should be surface-treated to increase thermodynamic miscibility and interface bonding strength before being used as reinforcement materials in polymer composites. In addition, surface modification is a required step that is rapidly becoming a major research area, particularly in the field of natural fiber modification. Surface modification methods such as silane treatment, radiation and discharge treatments, benzoylation treatment, peroxide treatment, use of maleate coupling agents, alkali treatment or mercerization and acetylation treatment have all been widely used so far and have the ability to improve the properties of natural fibers and bio-composites [55]. Figure 2 summarize the surface modification technique for cellulose-based fillers.

One disadvantage of cellulose nanocrystal (CNC), which is isolated and extracted from natural resources via hydrochloric acid (HCl) hydrolysis, is that it tends to aggregate, making it difficult to disperse in common solvents, which might disrupt processing. Dispersibility is generally determined by the surface functioning, aspect ratio and solvent’s capacity to disrupt the hydrogen bond interaction. CNC dispersion is commonly aided by the addition of negatively charged groups such as sulphate ester and carboxylic acid [20]. Due to the presence of the hydroxyl group of the NC surface, it has a significant agglomeration tendency, resulting in bigger particles with poor dispersion. To solve this problem, surface modification can boost the hydrophobicity and electrostatic stability of the NCs. Solvent exchange, physical changes and chemical modifications are the three basic types of NCs surface modification.

The most common modifications are physical and chemical, which are based on the addition of polymers or surfactants to the surface of the NCs. The chemical modification is the most common, characterized by the covalent attachment of a polymer grafting onto nanocellulose surface. This approach is effective at changing the polarity of nanoparticles, which offers a wide range of applications. Silylation, esterification, amidation, acetylation and carbanylation are the most commonly used technique in chemical modifications. Physical modification is a straightforward method that avoids the formation of covalent bonds, while ensuring cellulose stability. These techniques change the fibers’ dispersion and increase their hydrophobicity. Moreover, after nanocellulose isolation, most modification activities are carried out in aqueous solution, which requires more energy and time. In situ modification has been investigated by a few authors, while most modification is still conducted in aqueous solution to achieve a stable suspension [37].

CNC extracted from miscanthus gigantus (MxG) were modified by functionalized the CNC extracted with hydrochloric acid with 2,2,6,6-tetramethylpiperidinyloxy (TEMPO) oxidation to improve dispersibility of the CNC. The schematic reaction of TEMPO oxidation is shown in Figure 3. The hydrolysis of cellulose with sulfuric acid (H_2_SO_4_) also led to the formation of negatively charged sulphate ester groups on the surface of CNC. From these two reactions, it found that the crystallinity of the CNC also increased. In the dispersibility test, CNC modified with TEMPO oxidization improved dispersibility by being able to be held up to 7 days in water. Similar observations are also found on CNC hydrolyzed with (H_2_SO_4_) with the more viscous sample. Thermal degradation of the carboxylic treated MxG showed lower degradation temperature with higher weight residual loss, compared to both acid-hydrolyzed samples (H_2_SO_4_ and HCl). From the TEM results, the acid hydrolysis sample of the MxG shows aggregate, which makes it difficult to determine the dimension. The dimension of treated CNC is easier to measure since it can be dispersed in solution. It was found that the CNC has a ribbon-like shaped cross-section aspect ratio of the TEMPO-CNC and sulfuric acid-CNC is 69 and 63, respectively [39].

Another option for surface modification by chemical method is the use of anionic surfactant. The modification is conducted during the isolation process, which known as in situ ball milling modification. The raw material, which is soybean straw (SS), was first pre-treated with sodium hydroxide and bleaching process to remove non-cellulosic components. The surface modification takes place by using a mechanical method which is using ball milling with the addition of surfactants, which is condensed sodium alkyl naphthalene sulfonate (Surfom WG 8168). Fourier transform infrared (FTIR) results show the changes in the spectral changes of the cellulose side groups, which are attributed to the active surfactant groups. These changes indicate physical interactions between the compounds. Chemical bonds change the functional groups present on the material surface, while the physical interactions between the materials are the result of an electrical approximation between them, with no substitution of functional groups. The minor spectrum fluctuations are caused by conformational changes in the molecules as a result of their approximation, which alters the chain’s vibration modes.

X-ray photon spectroscopy (XPS) results show that the decrease in carbon and oxygen, together with the presence of Na and S, confirms the presence of surfactant on the cellulose surface. The milling time influenced the size distribution and as the longer milling time decreased the nanoparticle sizes (9 and 12 h). The in-situ modification generated the electrostatic interactions between the NC and the surfactant, which decreased the NC dimensions. From the TEM image, it can be observed that the modification process affects the morphology of the NC, where unmodified NCs showed a nanofibrillar shape with a diameter of ~400 nm, while in-situ modified NCs showed smaller particle sizes, nanocrystals shape and diameters between 100 and 150 nm. This indicates the efficiency of surfactant stabilization and electrostatic repulsions. It also suggests that the surfactant alters the milling process, making it more efficient. The surface-modified NC showed a similar crystalline structure than the unmodified NC and good thermal stability, expanding the applicability of nanocellulose in areas such as emulsifiers, coatings or fillers in polymer nanocomposites [37].

Esterification is the chemical modification technique that adjusts the hydrophobic/hydrophilic balance in cellulose, which improves its solubility, augmenting dispersion and interfacial adhesion. Surface modification based on the esterification method is explained in the project conducted by Singh et al. [47]; based on their study, the surface modification of cellulose isolated from wheat straw was conducted by using indirect esterification methods using propionic anhydride. The reaction mechanism of surface modification of nanofibrils is a two-stage process, an indirect method where the first involves the esterification of the primary and secondary hydroxyl groups of cellulosic surface present in MFCs followed by the isolation of surface-modified NFCs using mechanical disintegration.

The usage of propionic acid is found to have better dimensional stability in the resultant fiber and composites. The indirect method limits the reaction materials on the exposed surface of nanofibrils without causing any destruction in the fiber’s inner surface. Micro-fibrillated cellulose was soaked in a reaction medium of toluene, pyridine and H_2_SO_4_. Where toluene acts to restrict the swelling of the fiber and does not allow the reactants to enter the bulk site of fiber, pyridine is used to increase number of accessible reactive hydroxyl sites and enhance acetylation rate and H_2_SO_4_ as a catalyst. The presence of new bands on the modified MFC confirms the formation of ester binds with the free hydroxyl group present on the exposed surface fibers. The influence of propionylation on the morphology of modified MFCs was demonstrated by SEM examination. The surface of the steam exploded and the hydrolyzed MFCs of wheat straw are cleaner and smoother, but the surface of propionylated MFCs is rougher. Propynated groups on the surface are responsible for the roughness. From the TEM image, it was found that the high shear action of the homogenizer at a set time interval caused MFCs to disintegrate into fine NFCs and showed that the diameter of the nanofibrils in all samples was less than 100 nm. The tendency for treated samples to agglomerate is relatively low, showing that surface modification confers good hydrophobicity.

As a result of the indirect approach of propionylation, the hydrophobic character of the fiber surface significantly improved. With the increasing reaction time and temperature, the contact angle of the water drop rose. The rate of diffusion of the reactants and the rate of the reaction determined the course of the esterification reaction on the cellulose surface. The enhanced hydrophobicity of the nanofibrils has an impact on the thermal properties. At 100 °C, there was a decrease in the percentage weight loss after the change, which is linked to the evaporation of adsorbed water. The changed nanofibrils are less stable at higher temperatures, as observed by thermogravimetric analysis (TGA) and derivative thermogravimetric (DTG) plots. This phenomenon could be owing to the introduction of ester bonds into the cellulose structure or it could be related to a modest drop in crystallinity in the samples obtained after esterification.

In the study conducted by Barbosa et al. [35], microcrystalline cellulose (MCC) first underwent acid hydrolyzed with different type of acid which are sulfuric acid and hydrochloric acid. The sample was then acetylated using acetic anhydride in a homogeneous system and by using sulfuric acid as catalyst. From the result, the sulfuric acid that hydrolyzed sample showed smaller particle size compared to the hydrochloric acid sample. However, the thermal stability is lower for the sulfuric sample. This happens due to the presence of the sulphate ester group. From the morphology test, it was found that, after acetylation, the agglomeration is improved by observing the uniform size that obtained. After modification with acetic anhydride, the zeta potential of the sample was found, showing changes in the surface charge. The high negative charges results show the electrostatic stabilization of the sulfuric sample due to the sulphate ester group, which makes it stable for a longer time. The introduction of the acetyl group reduces the electrostatic stabilization due to the acetyl group, which diminishes the free surface charges of OH group and acts as a steric stabilization mechanism. This is supported by the FTIR, XPS and NMR analysis; the results show that the replacement of the OH group by acetyl group and has a high degree of substitution. The thermal stability of the sulfuric acid sample after acetylation also improved.

Other chemical modifications were conducted by using silylation techniques. The silylation technique, also known as silane grafting, is where the cellulose was introduced with hydrophobic alkyl groups such as chlorosilane. In the work by Khanjanzadeh et al. [56], a silylation technique was reported which modified CNC using the direct method and without uses of hazardous solvent. In their work, CNC was directly mixed with 3-aminopropyltriethoxysilane (APTES) solution at pH 4 and stirred. The sample was then centrifuged to collect the precipitate. In their result, the APTES-treated (ATR) sample shows the presence of N–H bonding which shows that the functional group is successfully added in the CNC. This is supported by the EDX and XPS results, which shows the presence of Si and N, which are attributed to APTES. The XPS detailed the increases of C, N and SI and reduction in O group due to the attenuation of oxygen from the cellulose surface, where C increased due to the presence of the propyl group from APTES. Nuclear magnetic resonance (NMR) results also show the presence of a new peak representing the resonance of an aminopropyl group of silanes coupling agent grafted into CNC. The crystallinity of the CNC showed no effect after modification, indicating that the crystalline structure of CNC was preserved and the atomic force microscope (AFM) shows that the modification does not impose any changes of the particle size and shape. While TGA results showed an increase in thermal stability after modification which, due to the bonding between APTES and CNS and the residue, are increasing due to siloxy moieties.

Jin et al. [57] also proposed the use 3-aminopropyltriethoxysilane (KH-550). Their work also used NCC, which is isolated using sulfuric acid and using the direct method for the surface modification of NCC. They proposed that, under acidic condition, the KH-550 form -Si-OH, which then formed hydrogen bonds with the hydroxyl group of the NCC or formed an ether bond through the dehydration or condensation process. The proposed alkylation mechanism is shown in Figure 4. The SEM result showed that SNCC has the image of the clubbed fibril network with a fuzzy boundary. The FTIR shows that the SNCC has a sharper intensity compared to NCC, which is due to the formation of hydrogen bonds. Other than that, the 1000–1100 band become broader and more complex, presumably due to the presence of the C-Si-O and Si-O-Si bonds. The thermal stability of the NCC also improved after treatment, suggesting that this is due to the hydrogen bonding of the NCC and the silane. Char residue also increased due to the presence of the alkoxysilane group. However, the crystallinity of the NCC after treatment was reduced. This is one of its more important properties, since the crystallinity may impact the final results of the composite. The reduced SNCC result is due to NCC being grafted with amorphous silane.

A comparison is made between two different methods of silylation using vapor technique and liquid technique [58]. The vapor technique is conducted by introducing the cellulose fiber with hexamethyldisilazane (HDMS) under the chemical vapor condition, while the liquid technique is conducted by soaking the CF in HDMS liquid. During silylation, the hydroxy group is converted to trimethylsiloxy group, which imparts hydrophobic properties. The FTIR shows the presence of a new peak at 1254 and 845, which, due to Si-C stretching, indicates that chemical moieties with trimethylsilyl appear on the surface of CF. Under SEM morphology, after modification, the surface of the fiber becomes smoother, especially the vapor techniques. The EDX shows the presence of the Si group on the modified CF, with vapor showing a higher value compared to the liquid. This was supported by the XPS result, with similar findings. XPS also shows a reduction in O/C ratio, which indicates a higher degree of silylation in the vapor method. The degree of surface substitution is higher in vapor than in the liquid treatment, which shows more hydrophobic wetting behavior and can be proved by the water contact angle result. Vapor treatment has higher water contact angle value compared to liquid treatment. This can be concluded that vapor technique has a higher functionalization degree of cellulose fiber surface which resulting more hydrophobic wetting behavior.

Cichosz et al. [59] investigated the effect of hybrid surface modification by combining the solvent exchange method and chemical modification and observed the effect of drying and not drying the cellulose before grafting. The grafting of cellulose was conducted by incorporating with maleic anhydride (MA). All the cellulose fibers show a decrease in moisture content after modifications. From the gathered data, it can be noted that the employment of ethanol highly contributed to the lowering of the water absorption. The cellulose, which was not dried before being grafted with MA, is more prone to grafting compared to a dried sample. All the samples that were treated with MA had lower thermal stability. The obtained results show that the cellulose fiber has a sufficient hydrophobilization level, suggesting that it is a promising method to modify the cellulose fiber.

## 3. Preparation and Characterization of Latex-Based Waste Natural Filler Composite

Metroxylan sagu pith waste (MSPW) is utilized as a new biodegradable filler in NR latex films. MSPW is a residue from the starch extraction process. It contains a high number of sago starch granules and high cellulose: up to 60%. The obtained results from incorporating MSPW with NR latex showed a reduction in tensile strength due to lignin, which hinder the formation of chains between rubber particles and creates distance between NR latex colloidal particles. A decreasing trend was also noted at the elongation at the break results and the modulus, due to the interparticle integration effect, which decreases the ability of the NR latex films to retain their stiffness. However, the tear strength increased with the addition of MSPW, due to the ability of the particle to deviate cracks, indicating a good cementing mechanism in NR latex films [12].

The presence of the glucopyranosyl group of starch, which consists of three hydroxyl groups, tends to form a strong hydrogen bond which results in filler agglomeration when incorporated with the XNBR latex. To solve this problem, amino-functional starch was produced by grafting the acrylonitrile monomer on the starch. This modification also helps increase the homogeneity of the filler dispersion in latex compound. The addition of the ANS, however, reduces the mechanical properties of the ANS/XNBR composites. This occurs due to the reduction in crosslink density between the rubber molecules, which was hindered by the formation of the ANS rich region. As biodegradation continues, further reductions in the mechanical properties are recorded due to the microbial activity in the ANS rich region, which were spotted in the SEM and optical micrograph. The formation of micro-voids was also detected in the XNBR latex films during biodegradation process. It was proved by the FTIR analysis that the reduction in starch aliphatic chain at peak 1607 cm^−1^ and stretching of N-H bond at peak 3302 cm^−1^ can be observed [11].

Another study was carried out using the acid hydrolyzed method in order to improve the mechanical properties of latex-sago starch composite, [60]. This method was able to induce the formation of the sulphate ester group on the surface of starch, which can increase the interaction between the rubber and starch. The XNBR latex mixed the compounding ingredients and acid hydrolyzed sago starch and formed into films using the coagulant dipping technique. AHSS-latex composite showed that the swelling percentage and mechanical properties strength improved compared to native sago starch, due to the increase of surface activity from smaller particles and the presence of sulphate ester, which creates the C-O-C bond, hence improving the rubber–filler interactions. The mass loss in AHSS-XNBR latex composite shows the highest value due to the loss of the amorphous region after acid hydrolysis, which led to microorganism attack on rubber and glycosidic chains.

Un-crosslinked NBR latex, which was added with nanocellulose from cotton seed linter pulp (CN), shows an increase in its viscosity, indicating the strong reaction between the CN and NBR chains. The foamed rubber produced has a smaller size, which provides better mechanical properties. Increasing the content of CN showed an improvement in tensile strength, which increased from 3.72 MPa to 6.54 MPa. The increment of tensile strength can be justified due to good interaction between CN and NBR matrix. This is supported through the SEM image shown in Figure 5; here, it can be observed that the CN was embedded in NBR matrix, which indicates a good interfacial adhesion. CN functioned as a physical crosslinking point with NBR chains and created hydrogen bonding and, hence, enhancing the crosslinked network of the composites. The possible scheme of the crosslink network between the NBR and CN is shown in Figure 6. CN also serves as a tensile stress carrier by transferring stress and prevent cracking and flaws from enlarging [61].

The incorporation of the nanocellulose fiber (CNF) from coconut spathe with NR latex was studied by Gopalakrishnan et al. [21]. The SEM result of the nanocellulose isolation showed that the diameter of nanocellulose is 40–50 nm; this is supported by the TEM results, where the diameter is around 30–60 nm. The optimum cure time of the NC/NR composite decreased with the increase of CNF content. Tensile, tear, modulus at 300 and hardness increase with increase of CNF, which indicates the high reinforcement of an NR matrix by well dispersed CNF. However, elongation at the break decreased with the increase of CNF, due to strong interaction between the rubber and filler, which increases its ability to resist the deformation of the NR chains. On the fracture surface, CNF was found to be embedded in the matrix, showing good compatibility between fiber and matrix with the presence of bonding agents. There were no voids or pull-out crack observed, which indicates good interfacial adhesion. The thermal properties of the maximum degradation remain the same, as suggested: that the thermal stability was not affected by the addition of CNF. The DMA modulus is directly proportional to the elastic nature of the composite. The modulus increased with the increase in CNF loading at low strain level, due to hydrodynamic reinforcement and the interactions of the filler–filler and filler–polymer. Solvent resistance towards toluene uptake decreased with higher loading CNF; this is due to the increase in the interfacial reaction between the NR chains and cellulose in the presence of the bonding agent, which increases the crosslinking point, hence preventing penetration of the solvent into rubber matrix.

Kenaf bast and kenaf core are used to determine their effect toward incorporation with NR latex. Kenaf/NR composite was prepared using the Dunlop method [62]. From the results, it can be observed that the tensile strength reduced with the increase in filler for both types of filler used. This happened due to kenaf core form agglomeration and led to a higher stress concentration, while kenaf bast has a larger pore size and is less interconnected with the matrix. Elongation at the break was also found to decrease with the increase in loading, due to the increase in the rigidity of the foam. The high loading fillers resulted in an NR latex foam with a higher stiffness, where the kenaf aggregates and stack acts as a filler network. Kenaf bast/NR foam has a higher stiffness than kenaf core due to its high cellulose content. The highly crystalline compact structure promotes higher stiffness. From the SEM image, the kenaf core has an irregular and particulate shape, and a tendency to agglomerate, making it less interconnected with rubber; meanwhile, bast has a fibrous shape and a higher length to cause the physical interaction between the filler and rubber. As incorporated with NR, the kenaf core forms an agglomeration, which forms a weak interaction between rubber–filler. Meanwhile bast is in a fibrous shape, which tends to cause a strong adhesion with rubber.

In the study on the effect of oxidization and hydrolyzation of corn starch, it can be noted that the particle size of the cornstarch was reduced by hydrolysis under alkali conditions at elevated temperature and high shearing force [63]. This can be explained by both the amide bond in protein and the ether bond in starch, which can be hydrolyzed under acid and base conditions in order to break up the polymer into smaller segment. The treatment shows that the CF has a size reduction of 33 times. The decrease in particle size helps improve the tensile strength, modulus and toughness, but there is a reduction in the elongation at the break. This occurs due to the increase in the filler–matrix contact area and an increase in the number of particles for filler network formation. Since corn flour has a greater modulus than NR, the rigidity of the connected article network penetrating throughout the NR matrix contributes to an increase in the modulus. The agglomeration effect can be explained through the Kraus plots in Figure 7. As the filler content of hydrolyzed CF composite was increased to 20%, it caused agglomeration, which led to the reduction of the restriction of polymer motion and, hence, increased the swelling ratio.

The hydrolysis of protein in the corn flour to generate negative charges was insufficient to cause particle–particle repulsion and reduce particle agglomeration at higher filler concentrations. The oxidation of corn flour converts hydroxyl functional groups in cornstarch to carboxylic acid functional groups, which are negatively charged under alkali conditions. The particles repel each other and prevent the formation of a particle network because of the increase in the negative charges on the surface of the corn flour particles. The composites filled with CF of low-level oxidation in the Kraus plot have a negative slope of the swelling behavior, which indicates that charge repulsion between particles prevented its agglomeration at the higher filler fractions. Lower crosslink density caused by high levels of oxidation resulted in poor tensile properties. As there are more COOH functional groups on the CF particle surface, the sulfur crosslink is retarded. A similar process uses a traditional retarder containing carboxylic acid groups, which interferes with the accelerator’s activity and reduces the cure time. The oxidized CF has a significantly lower viscosity, greater surface area and lower reinforcement effect than the hydrolyzed CF. Consequently, the dispersion viscosity was reduced [63].

In another work by Ab Rahman et al. [6], NR latex was incorporated with sago starch (SS) to observe the impact of dispersing agent towards the properties and biodegradability of NR latex films. The dispersing agents that were incorporated with sago starch were sodium alginate (SA), carboxymethyl cellulose (CMC) and anchoid (A). From the result, the particle size of SA/SS dispersion shows the smallest particle size and has the most stable result in zeta potential, which indicate good compatibility with sago starch. As the dispersion is incorporated with NR latex, the swelling percentage of NR/SA/SS decreases, compared to anchoid; this due to the increase in the rubber–filler interaction between the fillers. High crosslink density for NR/SA/SS was observed due to the increase in the interaction between sago and alginate to form a better filler–rubber interaction. The molecular entanglement in the rubber–filler can overcome the disadvantage of poor adhesion between rubber and sago starch. The presence of physical entanglement restricts the polymer chain mobility and reduces the penetration of the solvent into the NR latex film, which results in decreased of the swelling index.

The tensile strength of the unfilled is the highest; however, SS, which uses anchoid and SA as dispersing agents, has a tensile strength that meets the ASTM requirements. NR/CMC/SS has lowest tensile strength due to bigger particle size, which acts as stress concentration factor and, thus, reduces its ability to withstand further elongation. Larger and incompatible filler particles diminish or eliminate the ability to strain-induce crystallization, in which the particles physically disrupt the development of the crystalline lattice. The tear strength also showed unfilled NR film as the highest. NR/SA/SS has a high tear strength, compared to NR/CMC/SS; due to its smaller particle size and the presence of filler, it could resist crack growth during tear deformation. NR/SA/SS has a high tensile modulus due to SA, meaning that it can make a good stabilizer in the NR latex films, which enables the filler able to disperse well in the latex and reduce the mobility polymer chain mobility and, hence, increase the rigidity. From the visual analysis of degradation, it can be clearly observed that NR/SA/SS film shows more signs of biodegradation, with the formation of more yellow marks on the latex film; this indicates that the SA/SS disperses well in NR latex films and, hence, increases the rate of degradation [6].

NR latex was added to corn derivative bio-filler in order to study its impact on the properties, including biodegradation. The addition of corn starch and corn flour generally reduce the tensile strength of the film. From these three corn derivatives, corn grain (GC) has the optimum filer composition and highest compatibility with NR latex, which results in better physical and mechanical properties, which are low density, moderate hardness, elongation at break and higher tensile strength. At higher CG loading, the SEM confirmed the occurrence of biodegradation with the observation of mycelia forming microbial colonies at the composite surface. In addition, FTIR analysis proved that there are increases of aldehyde and ketone groups during soil burial, which indicate the progression of biodegradation. Another sign of degradation was also the detection of C-O-C and C-O bonds of primary alcohols, reducing of the carbon double bond and widening the epoxide group peak. Although higher CG loadings are better for achieving a higher degree of biodegradability, mechanical characteristics are drastically reduced. CG loadings greater than 20 phr of tensile retention are reduced, implying the rapid loss of initial tensile strength under ageing circumstances at larger biofiller loadings. As a result, the most ideal CG loading in NR latex films was determined to be 20 phr for the manufacture of biodegradable NR gloves, in line with ASTM D3578, which reported a biodegradability of roughly 50% after 15 weeks of soil burial [64].

## 4. Potential Applications

Cellulose can be widely applied in industry. Previously, cellulose was mostly used in the textile, paper and fine chemical industries; however, recently, cellulose has gained more interest in other applications, including the food, agriculture, pharmaceutical, medical, environmental and energy industries [30].

The incorporation of the cellulose nanofiber (CNF) with graphene (GNS) and NR latex creates a composite that is suitable for flexible display, sensor, biomedical and energy storage applications. The incorporation of CNF in GNS helps to distribute GNS and form a multi-layer crosslinked network structure. Except the increase in the maximum stress and elongation at the break of the composite, the CNF also helps improve the conductivity of the composites. The addition of CNF reduces the percolation threshold of the composite. The addition of CNF lowered the composite’s percolation threshold. Due to the high aspect ratio of the filler and its strong interfacial bonding, the multi-layered three-dimensional conductive network increased the electrical conductivity and overlaps easily in order to form a conductive network [65].

Hydrogel is hydrophilic polymer which consists of the 3-dimensional network. It has the capability to retain large amounts of water or biological fluids without being dissolved. Thanks to their features, they have recently gained the interest of numerous researchers in a variety of sectors, including drug delivery, wastewater treatment and agriculture. Most of the hydrogels on the market are synthesized from synthetic polymers, due to their excellent physical, chemical and mechanical properties. However, synthetic hydrogels have a high manufacturing cost and are non-renewable, difficult to decompose and not environmentally friendly. To solve these limitations, the hydrogels derived from natural polymers are very appealing solutions due to their good biodegradability, environmentally friendly nature and low production cost. Tanan et al. [66] conducted a study on the hydrogel of cassava starch-g-polyacrylic acid/natural rubber/polyvinyl alcohol (PVA) blend. The hydrogel was synthesized by aqueous solution polymerization. The semi-interpreting network is introduced by grafting cassava starch with acrylic acid and then mixing this with the NR/PVA blend; next, casting is conducted to obtain the film. The grafting of cassava starch results in the growth of the polyacrylic chain, which is then semi-interpenetrated with a PVANR solution to enhance the mechanical strength and flexibility of final products. From their research, it was found that the hydrogel exhibited an excellent water retention capacity, able to be reusable and have a high level of biodegradation. This property is a good indicator that it has a potential application in agriculture and horticulture. The improvement in its mechanical strength and thermal stability can be related to the H-bonding interaction between the OH group of PVA and the hydrophilic group on the network of grafted cassava.

Chitosan incorporated with epoxidized NR latex (ENR) is currently one of the most promising composites, which benefits the agricultural field. The bio-composite is used as a carrier in the slow release of fertilizer. The development of the slow-release copper compound is based on the entrapment of chitosan in ENR. This method will allow a matrix to be steadily altered with water-soluble polymer and released copper as a micronutrient in the fertilizer. In the slow released fertilizer, it is important to determine the effects of absorption, desorption and degradability. From the results, it can be observed that, as the chitosan loading increased, the absorption rate also increased due to the increase in the amino group, which creates an adsorption-active site. In addition to that, by its combination with ENR latex, chitosan altered its 3-dimensional structure and enhanced the contact with copper ion. The presence of the void from the SEM image also increases the adsorption site. The desorption rate of the bio-composite also depends on the chitosan loading. From kinetic study, it has been found that the desorption rate also depends on the chitosan content, where, with the increasing of chitosan content, the desorption rate also increased. The incremental increase in chitosan content increases the porosity of the rubber. As the chitosan swells, it allows the ions to diffuse through the pore formed and, hence, increases the release rate of the ions. For the biodegradation study, it was found that the increase of chitosan increases the biodegradation rate of the bio-composite. This result is a good indicator that once the release of copper ion through diffusion is stopped, the release of ions will be followed by the biodegradation method [67].

Junior et al. [68] studied the effect of bleached paper kraft and latex incorporated with cement. The stress–strain curve of the composite explains that the composite has a strain-softening behavior after the cracking of the matrix. This explained that the force is being transferred from the matrix to the fiber and thus, enables the composite to have a ductile behavior. The addition of cellulose to the composite has the biggest impact on the flexural result of the composite. The fiber, which is well distributed in the composite, helps to control the crack propagation and absorbing fractional energy in the post-cracking. While latex content assists in the reduction in the void of composites, it also helps by increasing the adhesion and tenacity of the composite by reducing the detachment of the interface between the fiber and the matrix.

Waterborne latexes are widely used in industrial application, such as adhesives, coatings and paintings. The most important requirement of waterborne latexes is the appearance after the application. The appearance depends on some properties, including rheology, substrate geometry and process parameters. The most common defect of waterborne latexes was the formation of sag. Jiang et al. [69] suggest that the addition of cellulose to acrylic latex improved the sag resistance of the latex and is thus beneficial for the waterborne applications. Cellulose nanocrystal has been used as a rheology modifier, in which it can help improve the viscosity. From the research results, acrylic latex with cellulose provides a gel-like behavior, where it shows that, as the concentration increases, the elastic and viscous moduli also increase. The increase in gel strength is due to stronger network formation. The suspension shows shear thinning behavior for viscosity. In the application of waterborne latexes, the viscosity of the latex must be low enough for atomization during spraying and increased in viscosity to prevent sagging. Creep properties determined the sag resistance of the composites. The addition of cellulose shows lower creep compliance under constant stress and high recovery of the strain. With the increase of cellulose content, the creep compliance at the end of constant stress clearly decreased, which occurred due to the network formation of cellulose and particle–particle interaction. A slight increase in the tensile strength and tensile modulus and reduction in the tensile strain were observed from the acrylic latex cellulose film, which occur due to the percolation phenomena. The improvement in thermal stability is explained by the high char formation, which acts as layers against further thermal degradation. The addition of nano size cellulose and it being well dispersed in latex does not disturb the transmittance properties of the latex, where it still provides high transparency to the latex films.

Eslami et al. [70] discuss the effect of modified nanocellulose, which is suitable for the dipped latex product. In their work, cellulose nanocrystal was modified using the grafting method by adding polylactic acid (PLA). The grafting approach suggests that the partial substitution of the OH group with PLA and, hence, CNC will have a dual nature, with mutual dispersibility in aqueous media. In this work, CNC and modified CNC (mCNC) were added into polychloroprene (CR) latex composite. Both CNC and mCNC-CR latex composites show an increase tensile strength and tensile modulus. A slightly increased tensile strength and modulus of the mCNC-CR composite contributed to better interaction between the PLA graft and the CR. From the TEM result, the CNC-CR composite shows the polygonal structure of the CNC, which is believed to occur due to the CNC being trapped in between coalescing particle and does not diffuse in CR. This structure is considered a partially structured network and a slight agglomeration, which results in early percolation. The polygonal structure is reduced in the mCNC-CR composite, explaining that mCNC is well dispersed and able to interact with CR and, thus, is in good agreement with the tensile strength of the composites. However, the incorporation of CNC and mCNC in CR caused increased water permeability, with mCNC-CR increasing until it reached the percolation limit of 3wt%. With good tensile and tear properties, these composites have the potential for wider application in dipped goods, such as medical gloves, membranes, catheters and balloons.

## 5. Conclusions

Natural filler-based composites have been widely used in order to preserve nature. Owing to their good mechanical properties, such as high strength, stiffness, durability and cellulose, they have caught the attention of many researchers aiming to fully utilize the potential of this natural resource. However, in comparison to synthetic fibers, these natural cellulosic fibers have poor mechanical properties, due to poor compatibility with the composite matrix. They are hydrophilic, which means that they disperse in the presence of hydroxyl, and the polar groups make the cellulose hydrophilic, which tends to form a strong network and agglomerates, making it difficult to uniformly dispersed in the latex matrix. As a result, natural fibers should be surface treated to increase the thermodynamic miscibility and interface bonding strength before being used as reinforcement materials in polymer composites. Through surface modification, the hydrophobicity of the cellulose can be improved and helps improve the dispersibility of the cellulose in the latex matrix. After surface modification, the mechanical and thermal properties of the composite were improved, and at the same time, provide biodegradability to the final products. The improvement in the dispersibility and properties of waste/raw natural filler in latex-based composite provides a wide range of potential applications in industries including biodegradable gloves, flexible display, sensor, medical application, agriculture and the construction industry.

## Figures and Tables

**Figure 1 polymers-13-03600-f001:**
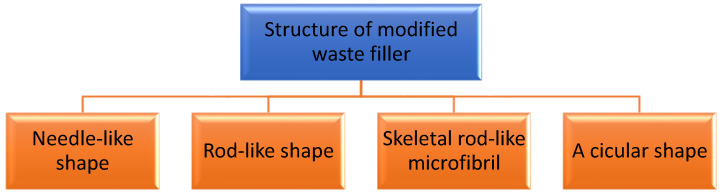
The different structure of modified waste/raw natural filler.

**Figure 2 polymers-13-03600-f002:**
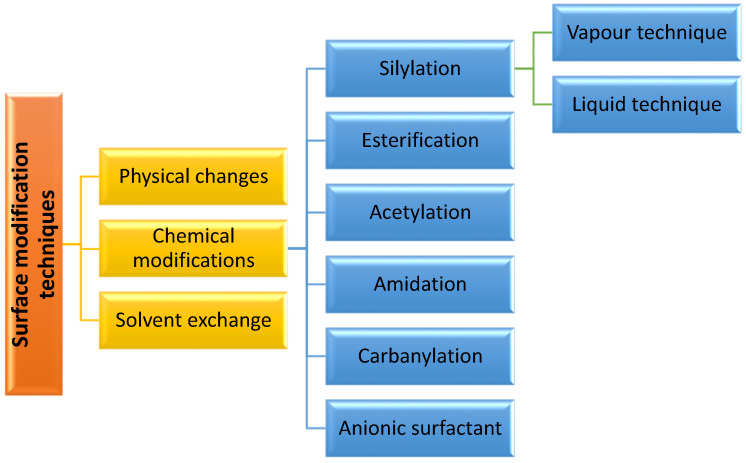
Summary on the surface modification technique for cellulose-based fillers.

**Figure 3 polymers-13-03600-f003:**
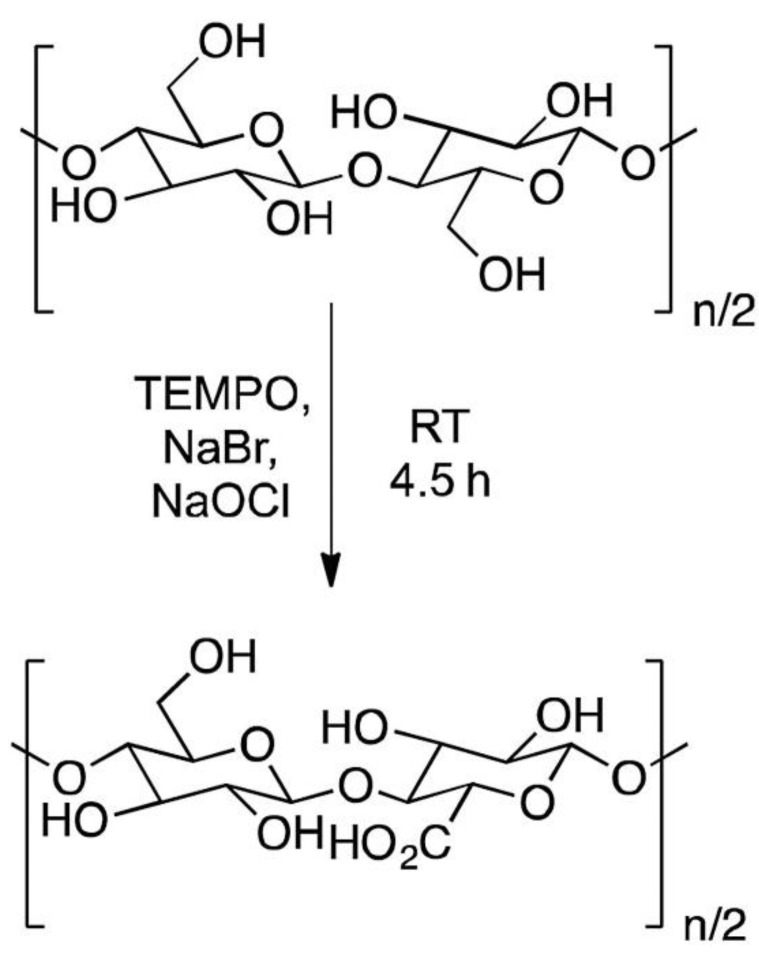
TEMPO oxidation of MxG-CNC to MxG-CNC-COOH.

**Figure 4 polymers-13-03600-f004:**
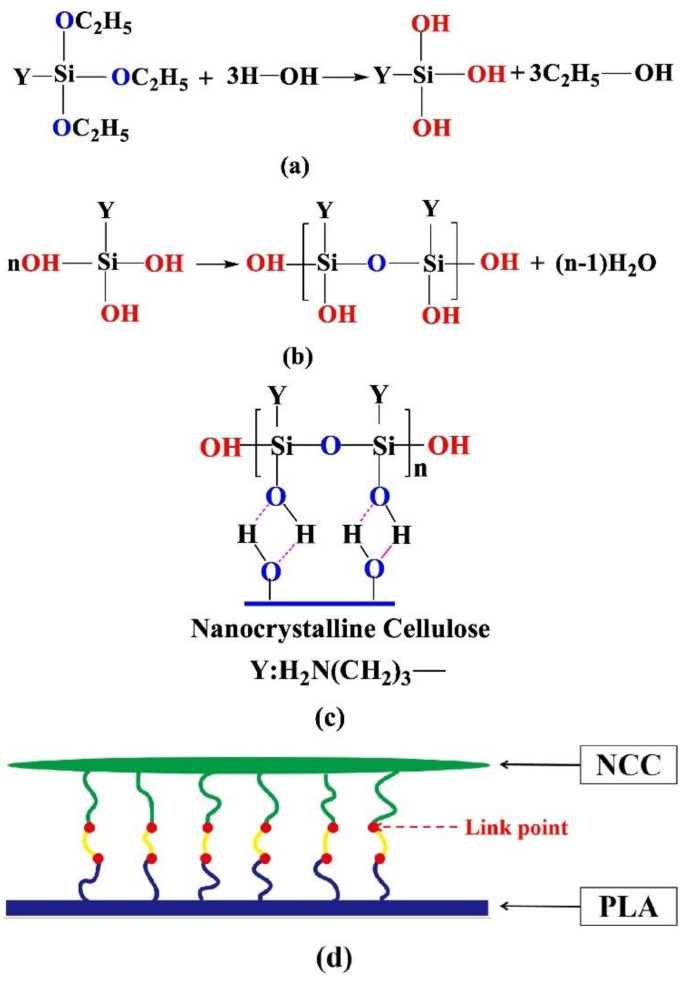
Proposed alkylation mechanism of NCC (**a**) Hydrolyzed of KH-550, (**b**) Oligomers formation, (**c**) Formation of hydrogen bond on surface of NCC (**d**) Coupling of KH-550 and NCC [57].

**Figure 5 polymers-13-03600-f005:**
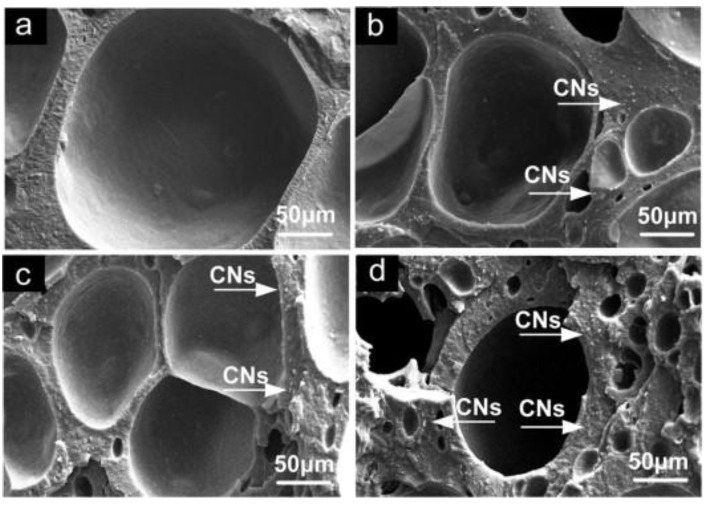
SEM images fractured surface of NBR/CNs nanocomposites (**a**) 3phr CNs, (**b**) 5 phr CNs, (**c**) 10 phr CNs and (**d**) 15 phr CNs [61].

**Figure 6 polymers-13-03600-f006:**
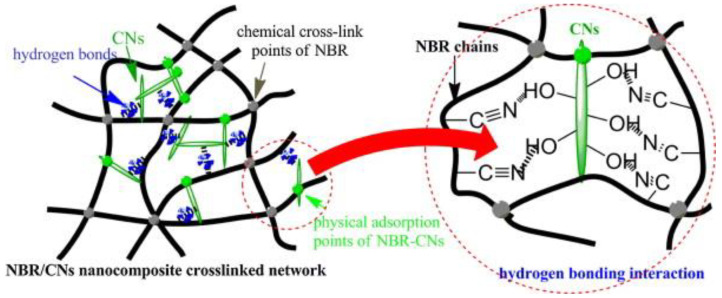
Possible scheme of crosslink network between NBR and CNs [61].

**Figure 7 polymers-13-03600-f007:**
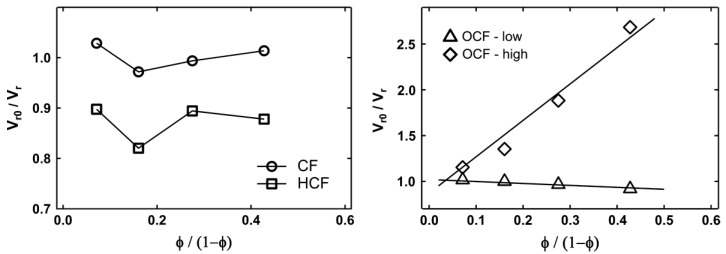
Kraus plot for corn flour (CF), hydrolyzed corn flour (HCF), oxidation corn flour-low oxidation level (OCF-low) and oxidation corn flour-high oxidation level (OCF-high) [63].

**Table 1 polymers-13-03600-t001:** Component in colloidal dispersion of NR latex [3].

Components	Percentage (*w*/*w*, %)
Rubber polymer (cis 1,4 polyisoprene)	25–35
Proteins	1.0–1.8
Carbohydrates	0.4–1.1
Neutral lipids	0.5–0.6
Inorganic components	0.4–0.6
Amino acids, amides, etc.	0.4
Water	50–70

**Table 2 polymers-13-03600-t002:** Properties of cellulose from different sources and extraction method.

Sources	Treatment	Structure	Size	Crystallinity, %	Reference
Date palm biomass waste	Acid hydrolysis and bleaching	Skeletal rod-like macro fibril structure,	180 µm	Rachis: 53.71 Leaflet: 50.66 Fiber: 52.43	[15]
Cocoa pod husk	Acid hydrolysis	Rod-like morphology,	10–60 nm 41–155 nm	67.60	[17]
Industrial waste cotton	Acid hydrolysis	Rod-like structure,	10 nm 190 nm	N/A	[18]
Soy hull	Acid hydrolysis	Long crystal,	49 ± 1.1 nm 503 ± 155 nm	80.4	[50]
Rubberwood (RW), Kenaf bast (KB)	Acid hydrolysis	Needle shape	Rubberwood: 33.54 nm Kenaf bast: 5.14 nm	RW:74.34 KB: 73.19	[24]
Jute fiber	Steam explosion and low acid hydrolysis	Nanocrystal	50 nm	82.22	[51]
Coconut spathe	Acid hydrolysis	Nanofiber	30–60 nm	N/A	[21]
Shrimp shell waste	Acid hydrolysis	Rod-like structure	25–32 nm 400 nm	86	[48]
Cotton, cotton stalk pulp	H_2_SO_4_ hydrolysis	Rod shape nanofiber	Cotton: 460 nm, 25 nm Cotton stalk: 100–850 nm, 25–100 nm	91, 86.3	[20]
Hemp fiber	H_2_SO_4_ hydrolysis	Rod shape structure	(160 ± 20) nm (4.5 ± 1) nm	N/A	[52]
Wood	Ethanol, peroxide and ultrasonic	Rod-like shape	1–9 nm, l < 500 nm	84.37	[53]
Barley straw and husk	NaOH and sulfuric acid	Acicular structure	L: (270 ± 40) nm W:(15 ± 5) nm	N/A	[49]
Grain straw	H_2_SO_4_ hydrolysis, homogenizer and ultrasonic	Rod shape	L:120–800 nm W:10–25 nm	28.6–41.4	[54]

## Data Availability

The data presented in this study are available on request from the corresponding author.

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
