# Peer review of "Waste Natural Polymers as Potential Fillers for Biodegradable Latex-Based Composites: A Review"

_polymers, 2021, doi:10.3390/polym13203600_

Round 1

Reviewer 1 Report

Reviewers' comments:

Manuscript Number: polymers-1408185

Full Title: Waste Natural Fillers as potential fillers for biodegradable latex- based composites: A Review.

Comments: 

The manuscript reported on Waste Natural Fillers as potential fillers for biodegradable latex- based composites: A Review. The manuscript needs a detailed editing. It cannot be recommended for publication in the present form. I hope the following points would be helpful for the authors.

- Some sentences need reconstruction and the level of English should be improved.

- In the Abstract, the authors need to improve.

- Keywords: add some more keywords.

- The introduction section should be improved; more related papers must be discussed and superiority, novelty, critical improvement in this study must be clarified.

- More recent literature need to be added (only 45 reference is not enough to review paper).

- Figure 3. Not clear make clear.

- Author should check sectional numbers throughout the manuscript (5. Potential applications………. To…….. 4. Potential applications; 6. Conclusions……… to…….. 5. Conclusions).

- Potential applications – should be improve.

- Conclusion should be concise.

- References: there are recent references in 2021 treating the same subject, you can use and make all references in same format for volume number, page numbers and journal name, because it is difficult to searching and reading.

- Several faults: are added or missing spaces between words: see manuscript file.

Author Response

Please see the attchment

Reviewer 2 Report

In this review, an attempt is made to discuss the main methods of processing composites based on natural cellulose fibers, which can improve their mechanical properties. It is shown, that these modifications not only improve the mechanical and thermal properties but also ensures the biodegradability of the final products. The overview is clear enough and well structured. However, Figure 3 is of very poor quality and should be improved. In general, the review is of great practical importance and can be recommended for publication.

Author Response

Thank you very much for the nice comment.

Figure 3 that very poor quality and should be improved - We already replace Figure 3 with more sharp images.

Round 2

Reviewer 1 Report

The authors revised the manuscript according to the reviewers' comments.